DATA RELEASE

# Annotation of glycolysis, gluconeogenesis, and trehaloneogenesis pathways provide insight into carbohydrate metabolism in the Asian citrus psyllid

Blessy Tamayo[1], Kyle Kercher[1], Chad Vosburg[1], Crissy Massimino[1], Margaryta R. Jernigan[1], Denisse L. Hasan[1], Douglas Harper[1], Anuja Mathew[1], Samuel Adkins[1], Teresa Shippy[2], Prashant S. Hosmani[3], Mirella Flores-Gonzalez[3], Naftali Panitz[3], Lukas A. Mueller[3], Wayne B. Hunter[4], Joshua B. Benoit[5], Susan J. Brown[2], Tom D'Elia[1] and Surya Saha[3,6,*]

1  Indian River State College, Fort Pierce, FL 34981, USA
2  Division of Biology, Kansas State University, Manhattan, KS 66506, USA
3  Boyce Thompson Institute Ithaca, NY 14853, USA
4  US Department of Agriculture-Agricultural Research Service (USDA-ARS), US Horticultural Research Laboratory, Fort Pierce, FL 34945, USA
5  Department of Biological Sciences, University of Cincinnati, Cincinnati, OH 45221, USA
6  Animal and Comparative Biomedical Sciences, University of Arizona, Tucson, AZ 85721, USA

## ABSTRACT

Citrus greening disease is caused by the pathogen *Candidatus* Liberibacter asiaticus and transmitted by the Asian citrus psyllid, *Diaphorina citri*. No curative treatment or significant prevention mechanism exists for this disease, which causes economic losses from reduced citrus production. A high-quality genome of *D. citri* is being manually annotated to provide accurate gene models to identify novel control targets and increase understanding of this pest. Here, we annotated 25 *D. citri* genes involved in glycolysis and gluconeogenesis, and seven in trehaloneogenesis. Comparative analysis showed that glycolysis genes in *D. citri* are highly conserved but copy numbers vary. Analysis of expression levels revealed upregulation of several enzymes in the glycolysis pathway in the thorax, consistent with the primary use of glucose by thoracic flight muscles. Manually annotating these core metabolic pathways provides accurate genomic foundation for developing gene-targeting therapeutics to control *D. citri*.

**Submitted:** 12 October 2021

\* Corresponding author. E-mail: suryasaha@cornell.edu

Preprint submitted at https://doi.org/10.1101/2021.10.11.463922

Included in the series: *Asian citrus psyllid community annotation* (https://doi.org/10.46471/GIGABYTE_SERIES_0001)

**Subjects**  Genetics and Genomics, Animal Genetics, Bioinformatics

## DATA DESCRIPTION

### Background

Huanglongbing (HLB), or citrus greening disease, is the biggest global threat to the citrus industry throughout the world [1]. The phloem-limited bacterial pathogen *Candidatus* Liberibacter asiaticus (*C*Las) is the causative agent of HLB. Upon infection of a citrus tree,

HLB causes development of small, bitter fruits, loss of tree vigor, fruit drop, and ultimately tree decline and death [1–4]. This bacterium is transmitted by the psyllid vector, *Diaphorina citri* (NCBI:txid121845), when feeding on citrus [5, 6]. Pesticide application to eliminate *D. citri* has been unsuccessful and no cure for HLB exists [7, 8]. To develop new psyllid control strategies, the International Psyllid Genome Consortium was established in 2009 [9] to provide the genome, transcriptome resources, and an official gene set of *D. citri* [10, 11]. A recent, nearly complete genome with significantly improved gene accuracy has been generated, providing a valuable dataset for the establishment of gene-targeted strategies to suppress psyllid populations (opensource: Diaci_v3.0, www.citrusgreening.org [12]; USDA-NIFA grant 2015-70016-23028). As part of this genome project, we manually annotated genes in critical pathways to provide the quality gene models required to design molecular therapeutics such as RNA interference (RNAi) [13–21], antisense oligonucleotides (ASO) [16, 20, 22] and gene editing (CRISPR) [23, 24]. Here, we examined *D. citri* orthologs associated with the critical metabolic pathways glycolysis, gluconeogenesis, and trehaloneogenesis.

## Context

A community-driven annotation strategy was used to identify and characterize the genes encoding enzymes involved in glycolysis, gluconeogenesis, and trehaloneogenesis (Figure 1).

Glycolysis is vital metabolic pathway in core energy processing reactions, and provides a source of metabolites for other biochemical processes. Insects utilize much glucose in flight muscles in the thorax [28]. Accordingly, the activities of glycolytic enzymes are increased in insect flight muscle compared with vertebrate muscle tissue [29]. Gluconeogenesis is essential in insects to maintain sugar homeostasis and serves as the initial step towards generating glucose disaccharide, also known as trehalose. Trehalose is the main circulating sugar in the insect hemolymph [30–32]. In trehaloneogenesis, glucose-6-phosphate is converted into trehalose by trehalose-6-phosphate synthase (TPS). Trehalase enzymes then degrade trehalose into two glucose molecules [33]. Genes involved in psyllid glycolysis, gluconeogenesis, and trehaloneogenesis have been targeted by several RNAi studies (Table 1) as a promising avenue for psyllid population suppression. In particular, one proof of concept experiment targeting trehalase led to the release of the first RNAi patent to control psyllid populations [49]. RNAi, as a biopesticide, and strategies for delivery and applications to target insect pests and viral pathogens have been thoroughly reviewed [50–54].

## METHODS

The *D. citri* genome was manually annotated through a collaborative community-driven strategy [11] with an undergraduate focus that allows specific students to focus on main gene sets [55]. Orthologous protein sequences for the glycolysis, gluconeogenesis, and trehaloneogenesis pathways were obtained from the National Center for Biotechnology Information (NCBI) protein database [56] and were used to BLAST the *D. citri* MCOT (Maker (RRID:SCR_005309), Cufflinks (RRID:SCR_014597), Oases (RRID:SCR_011896), and Trinity (RRID:SCR_013048)) protein database to find predicted protein models [25]. MCOT predicted protein models were used to search the *D. citri* genomes (versions 2.0 and 3.0) [55]. Regions of high sequence identity were manually curated in Apollo v2.1.0 (RRID:SCR_001936) using

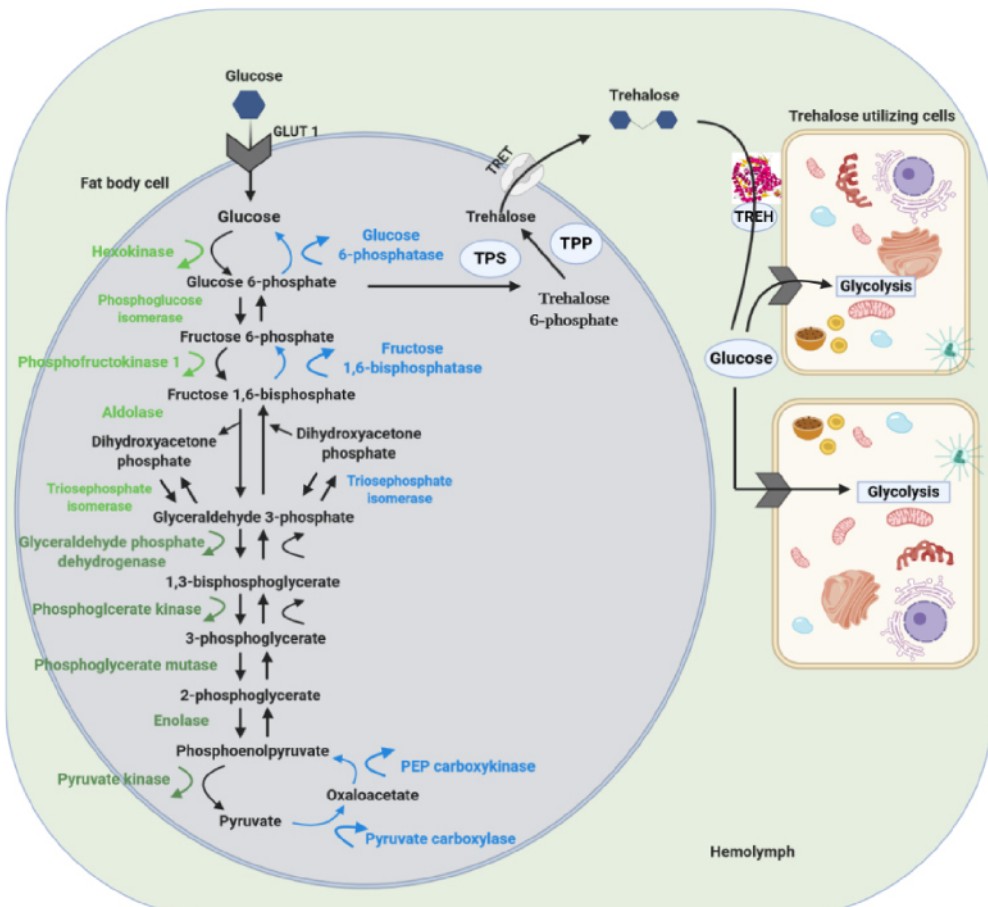

**Figure 1.** Overview of the glycolysis, gluconeogenesis, and trehaloneogenesis pathways. The pathway image shows the enzymes that produce and utilize glucose and trehalose in insects [25]. The glycolysis pathway comprises 10 enzymes that convert glucose into pyruvate as the final product. These are divided into the energy investment phase (light green) and the energy production phase (dark green). The gluconeogenesis pathway comprises eight enzymes (blue), with three being unique to the pathway that bypasses the irreversible reactions in glycolysis to convert non-carbohydrate molecules into glucose. The trehaloneogenesis pathway comprises three enzymes: trehalose-6-phosphate synthase (TPS), trehalose-6-phosphate phosphatase (TPP), and trehalase (TREH), as well as trehalose transporters (TRET) and glucose transporters (GLUT1). Image adapted from a diagram in [26] and created with BioRender.com [27].

*de novo* transcriptome, MCOT gene predictions, RNA-seq, Iso-seq, and ortholog data to support and evaluate gene structure (Table 2). The curated gene models were compared with other orthologous sequences, such as hemipterans, available through NCBI for accuracy. A more detailed description of the annotation workflow is available (Figure 2) [58].

Neighbor-joining phylogenetic trees of the annotated *hexokinase* gene models in *D. citri* and orthologous sequences were created with MEGA v7 (RRID:SCR_000667) using the MUSCLE (RRID:SCR_011812) multiple sequence alignment with *p*-distance for determining branch length and 1,000 bootstrap replicates [59].

Expression levels of the carbohydrate metabolism genes throughout different life stages (egg, nymph, and adult) in *C*Las infected and uninfected *D. citri* insects were collected from

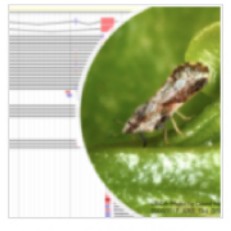

Annotating genes in Diaphorina citri genome
version 3

Teresa Shippy[1], S Miller[2], C Massimino[3], C Vosburg [Indian
River State College[4]. PS Hosmani[5]. M Flores-Gonzalez[5]. LA...
[1]Kansas State University, [2]Kansas State University, Allen
County Community College. [3]Indian River State College....

Dec 16, 2020

**Figure 2.** Protocol for *D. citri* genome community curation [58]. https://www.protocols.io/widgets/doi?uri=dx.doi.org/10.17504/protocols.io.bniimcce

**Table 1.** List of annotated genes in glycolysis, gluconeogenesis, and trehaloneogenesis.

| Genes | Organism | RNAi outcome | Reference |
|---|---|---|---|
| *Hexokinase* (*HK*) *Tc-HexA1* | *Tribolium castaneum* | *HexA1* role in glucose metabolism is essential during oogenesis and embryogenesis | [34] |
| *Aldolase UAS-Aldolase-RNAi* | *Drosophila melanogaster* | Knockdown in *Drosophila* neurons and glia resulted in reduced lifespan; essential in glia for neuronal maintenance | [35] |
| *Enolase a-enolase* | *Nilaparvata lugens* | Knockdown reduced egg production, offspring and hatching rate; mortality of adults was unaffected | [36] |
| *Pyruvate kinase* (*PYK*) *NlPYK* | *Nilaparvata lugens* | RNAi treatment including triazophos and *dsNlPYK* led to reduced ovarian protein content, ovarian and fat body soluble sugar contents, and fecundity | [37] |
| *Phosphoenolpyruvate carboxykinase* (*PEPCK*) | *Drosophila melanogaster* | Knockdown of two *PEPCK* mutant isoforms led to reduced circulating glycerol levels and reduced triglyceride levels in *pepck1* mutant flies | [38] |
| *Trehalose-6-phosphate synthase* (*TPS*) | *Diaphorina citri* | Knockdown of the *Trehalose- 6-phosphate synthase* gene using RNA interference inhibits synthesis of trehalose and increases lethality rate in Asian citrus psyllid | [39] |
| *Trehalose phosphate synthase* (*TPS*) *NlTPS* | *Nilaparvata lugens* | Feeding *N. lugens* larvae with *NlTPS* dsRNA led to disrupted expression and lethality | [40] |
| *Trehalose-6- phosphate synthases* | *Nilaparvata lugens* | Silencing of two TPS genes can lead to increased molting deformities and mortality rates leading to misregulation of chitin metabolism genes | [41] |
| *chitin synthase** | *Diaphorina citri* | Silencing of the chitin synthase gene is lethal to the Asian citrus psyllid | [42] |
| Five *trehalase* genes | *Tribolium castaneum* | Regulates gene expression of the chitin biosynthesis pathway | [43] |
| *Trehalase* genes (*TRE*) | *Nilaparvata lugens* | Wing bud chitin metabolism and its development | [44] |
| *Trehalase* | *Nilaparvata lugens* | Regulating the chitin metabolism pathway | [45] |
| *Muscle protein 20* ‡ | *Diaphorina citri* | Increases mortality to the Asian citrus psyllid | [46] |
| *Sucrose hydrolase* ‡ | *Diaphorina citri* | Causes nymph mortality and disturbs adult osmotic homeostasis | [47] |

List of annotated genes in glycolysis (*HK, aldolase, enolase, PYK*), gluconeogenesis (*PEPCK*), and trehaloneogenesis (*TPS* and *TREH*), with their corresponding RNAi studies and references. ‡ indicates that additional genes were added, but not annotated in *D. citri*, such as *muscle protein 20* and *sucrose hydrolase*. *indicates that the *chitin synthase* gene in the chitin synthesis pathway was also annotated in *D. citri* [48].

the Citrus Greening Expression Network (CGEN) [25] and visualized using Excel (RRID:SCR_016137) and the pheatmap package in R (RRID:SCR:_016418).

## DATA VALIDATION AND QUALITY CONTROL

There are four phases of the carbohydrate metabolism pathways in *D. citri*: the energy investment phase of glycolysis, the energy production phase of glycolysis, gluconeogenesis, and trehaloneogenesis. Enzymes involved in the breakdown and synthesis of glucose and trehalose were annotated in version 3.0 of the *D. citri* genome [57]. The following genes in the energy investment phase: *hexokinase* (*HK*), *phosphoglucose isomerase* (*PGI*), *phosphofructokinase* (*PFK*), *fructose-bisphosphate aldolase* (*aldolase*), *triosephosphate isomerase* (*TPI*), and in the energy production phase: *glyceraldehyde phosphate dehydrogenase* (*GAPDH*), *phosphoglycerate kinase* (*PGK*), *phosphoglycerate mutase* (*PGAM*),

**Table 2.** List of annotated *D. citri* models along with their evidence.

| GENE | Identifier | MCOT | *de novo* transcripts | Iso-seq | RNA-seq | Ortholog |
|---|---|---|---|---|---|---|
| **GLYCOLYSIS** | | | | | | |
| *Hexokinase type 2-1* | Dcitr03g04910.2.1 | x | x | x | x | |
| *Hexokinase type 2-2* | Dcitr03g19430.1.1 | x | x | x | x | x |
| *Hexokinase type 2-3* | Dcitr06g14200.1.1 | x | | x | x | |
| *Phosphoglucose isomerase* | Dcitr00g06460.1.1 | | x | x | x | x |
| *Glucose-6-phosphate 1-epimerase** | Dcitr13g02890.1.1 | x | x | x | x | x |
| *ATP Dependent 6-Phosphofructokinase RA* | Dcitr01g16570.1.1 | x | x | x | x | x |
| *ATP Dependent 6-Phosphofructokinase RB* | Dcitr01g16570.1.2 | x | x | x | x | x |
| *ATP Dependent 6-Phosphofructokinase RC* | Dcitr01g16570.1.3 | x | x | x | x | x |
| *Fructose-bisphosphate aldolase 1* | Dcitr04g02510.1.1 | x | x | x | x | x |
| *Fructose-bisphosphate aldolase 2* | Dcitr11g09140.1.1 | x | x | x | x | |
| *Triosephosphate isomerase* | Dcitr10g08030.1.1 | x | x | x | x | x |
| *Glyceraldehyde 3-phosphate dehydrogenase-like 1* | Dcitr10g11030.1.1 | x | | x | x | x |
| *Glyceraldehyde 3-phosphate dehydrogenase-like 2* | Dcitr01g03200.1.1 | | | x | x | |
| *Phosphoglycerate kinase* | Dcitr00g01740.1.1 | | | x | x | x |
| *Phosphoglycerate mutase 1* | Dcitr03g11640.1.1 | x | x | | x | |
| *Phosphoglycerate mutase 2* | Dcitr03g17850.1.1 | | | x | x | x |
| *Enolase* | Dcitr02g07600.1.1 | | | x | x | x |
| *Pyruvate kinase-like 1* | Dcitr07g06140.1.1 | x | x | x | x | x |
| *Pyruvate kinase-like 2* | Dcitr01g11190.1.1 | x | x | x | x | |
| *Phosphoglucomutase 1* | Dcitr05g09820.1.1 | x | | x | x | |
| *Phosphoglucomutase 2* | Dcitr02g10730.1.1 | | | x | x | x |
| **GLUCONEOGENESIS** | | | | | | |
| *Pyruvate carboxylase* | Dcitr08g01610.1.1 | x | x | x | x | x |
| *Phosphoenolpyruvate carboxykinase 1* | Dcitr05g10240.1.1 | x | x | x | x | |
| *Phosphoenolpyruvate carboxykinase 2* | Dcitr08g02760.1.1 | x | | x | x | x |
| *Aldose 1-epimerase 1** | Dcitr04g09830.1.1 | x | | x | x | x |
| *Aldose 1-epimerase 2** | Dcitr06g04430.1.1 | | | x | x | |
| *Fructose-1,6-bisphosphatase* | Dcitr11g08070.1.1 | | x | | x | x |
| **TREHALONEOGENESIS** | | | | | | |
| *Trehalose-6-phosphate synthase 1 (TPS 1)* | Dcitr02g17550.1.1 | x | | x | x | |
| *Trehalose-6-phosphate synthase 2 (TPS 2)* | Dcitr01g19625.1.2 | x | x | x | x | |
| *Trehalase 1A (TREH-1A) Isoform A* | Dcitr07g04030.1.1 | x | x | x | x | |
| *Trehalase 1B (TREH-1B) Isoform B* | Dcitr07g07175.1.2 | x | x | x | x | |
| *Trehalase 2 (TREH-2)* | Dcitr08g09220.1.1 | x | x | | x | |
| *Trehalose transporter 1 (TRET1) Isoform (TRET1A)* | Dcitr01g17710.1.1 | | x | x | x | |
| *Trehalose transporter 1 (TRET1) Isoform (TRET1B)* | Dcitr01g17715.1.2 | x | x | x | x | |
| *Trehalose transporter 2 (TRET2) Isoform (TRET2A)* | Dcitr00g03240.1.1 | x | x | x | x | |
| *Trehalose transporter 2 (TRET2) Isoform (TRET2C)* | Dcitr09g02310.1.3 | | | x | x | |
| *Glucose transporter (GLUT1)* | Dcitr05g13950.1.1 | x | x | x | x | x |

Each manually annotated gene in glycolysis, gluconeogenesis, and trehaloneogenesis associated with a *D. citri* identifier shows supporting evidence used in the curation of the gene model [57]. Evidence tracks are as follows: RNA-seq, long-read Iso-seq, MCOT, *de novo* assembled transcripts and orthologous proteins. A gene marked with an "x" within the table indicates that the gene model is supported by the evidence track. A gene followed by "*" indicates that it is involved in both glycolysis and gluconeogenesis.

*enolase*, and *pyruvate kinase* (*PYK*) were annotated. The annotated genes for gluconeogenesis are *pyruvate carboxylase* (*PC*), *phosphoenolpyruvate carboxykinase* (*PEPCK*), and *fructose 1,6-bisphosphatase* (*FBPase*). In trehaloneogenesis, *trehalose transporter 1* (*TRET1*) and *2* (*TRET2*), *glucose transporter 1* (*GLUT1*), and two gene models of both *trehalose-6-phosphate synthase* (*TPS*) and *trehalase* (*TREH*) were annotated. Gene expression datasets in CGEN were analyzed for potential differences, as expression patterns can provide insight into potential RNAi target candidates for molecular therapeutics (Table 1).



**Table 3.** Orthologs used in phylogenetic analysis and multiple sequence alignments.

| Protein | *Drosophila melanogaster* | *Tribolium castaneum* | *Apis mellifera* | *Acyrthosiphon pisum* | *Drosophila pseudoobscura* | *Halyomorpha halys* | *Nilaparvata lugens* |
|---|---|---|---|---|---|---|---|
| Hex-A | NP_001259384.1 | XP_008201714.1 | XP_006557646.1 | XP_003242238.1 | XP_001355083.1 | | XP_022204875.1 |
| | | XP_970645.1 | | XP_001952412.1 | | | |
| Hex-type 1 | | | | | | XP_014282249.1 | XP_022184109.1 |
| Hex-type 2 | | | | | | XP_014282721.1 | |
| Hex-C | NP_524674.1 | | | | XP_001360104.2 | | |
| Hex-t1 | NP_788744.1 | | | | XP_001359146.2 | | |
| Hex-t2 | NP_733151.2 | | | | XP_002137641.2 | | |

The accession numbers for *hexokinase* orthologs were obtained from the NCBI database.

Orthologous sequences from related insects and information about conserved motifs or domains were used to determine the final annotation. We used proteins from *Drosophila melanogaster* (*Dm*) [60], *Tribolium castaneum* (*Tc*) [61], *Apis mellifera* (*Am*) [62], *Acyrthosiphon pisum* (*Ap*) [63], *Nilaparvata lugens* (*Nl*) [64, 65], and *Halyomorpha halys* (*Hh*) [66]. Accession numbers are provided in Table 3.

### Energy investment phase of glycolysis

*HK* catalyzes the first step in glycolysis, utilizing adenosine triphosphate (ATP) to phosphorylate glucose, creating glucose-6-phosphate. Most insects have multiple *HK* genes and three copies of *HK* are present in the *D. citri* genome (Figure 3, Tables 2 and 4). In insect flight muscles, *HK* activity is inhibited by its product, glucose-6-phosphate, to initiate flight muscle activity [69]. *Drosophila melanogaster* has four duplicated *HK* genes, with *Hex-A* being the most conserved and essential flight muscle *HK* isozyme among *Drosophila* species [70, 71]. For *Diasporina citri*, one of the copies of *HK* type 2-2 (Dcitr03g19430.1.1) showed moderate expression in the male and female thorax. In contrast, another copy *HK* type 2-3 (Dcitr06g14200.1.1), was highly expressed in the adult gut and midgut compared with *HK* type 2-2 and its overall expression (Figure 4). *PGI* catalyzes the interconversion of glucose-6-phosphate and fructose-6-phosphate in the second step of glycolysis. Consistent with the gene copy number of *PGI* for orthologs in other insects, such as *D. melanogaster*, *Apis mellifera*, *Acyrthosiphon pisum*, and *Tribolium castaneum*, a single copy of *PGI* (Dcitr00g06460.1.1) was found. Expression for *PGI* is high in the male and female thorax (Figure 4).

*PFK*, which catalyzes the phosphorylation of fructose-6-phosphate using ATP to generate fructose-1,6-bisphosphate and adenosine diphosphate (ADP), is the key regulatory enzyme controlling glycolysis in insects, as it catalyzes a rate-determining reaction [76, 77]. One copy of *PFK* (Dcitr01g16570.1.1) was found and annotated in *D. citri* (Table 4). *Aldolase* catalyzes the fourth step, the reversible aldol cleavage of fructose-1,6-bisphosphate to form two trioses, glyceraldehyde-3-phosphate (GAP) and dihydroxyacetone phosphate (DHAP). Although most insects have a single copy of this gene, two well supported copies were found in *D. citri* (Table 4). One of the *aldolase* annotated copies, *fructose-bisphosphate aldolase 1*, (Dcitr04g02510.1.1) appears to have moderate expression in the male abdomen and terminal abdomen, and highest expression in the adult whole body (Figure 4). *TPI* catalyzes the fifth step, the reversible interconversion of DHAP and GAP. *TPI* is also important to sustain DHAP to maintain insect flight muscle activity [78]. *D. citri* contains a single copy of this gene (Dcitr10g08030.1.1), which is consistent with other insects (Table 4).



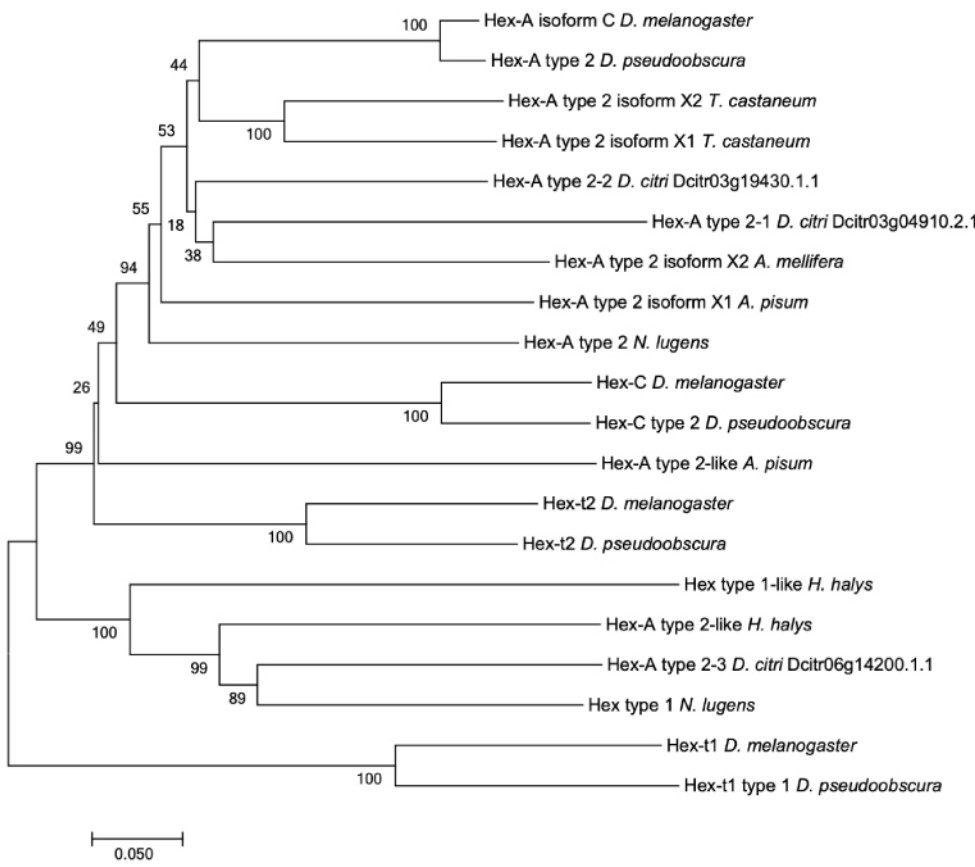

**Figure 3.** Phylogenetic analysis of *hexokinase* (*HK*). *Hexokinase* amino acid sequence of *D. citri* compared with sequences from other insects. MUSCLE multiple sequence alignments of *HK* in *D. citri* and orthologs were performed using MEGA7 and neighbor-joining phylogenetic trees were constructed with *p*-distance for determining evolutionary distance and 1000 bootstrapping replicates [59]. Accession numbers for the orthologous sequences used in phylogenetic analysis are in Table 3.

Expression of several of these genes in the investment phase was high in the male and female thorax, especially in *PFK* (Dcitr01g16570.1.1), *fructose-bisphosphate aldolase 2* (Dcitr11g09140.1.1), and *TPI* (Dcitr10g08030.1.1) (NCBI BioProject PRJNA448935) (Figure 4 and Table 3 in GigaDB [79]).

## Energy production phase of glycolysis

*GAPDH* catalyzes the reversible conversion of GAP to 1,3-bisphosphoglycerate during glycolysis. Two *GAPDH* genes were annotated in *D. citri* and the expression data for the two paralogs show that *GAPDH-like 1* (Dcitr10g11030.1.1) has higher expression in the male terminal abdomen and whole body and *GAPDH-like 2* (Dcitr01g03200.1.1) has higher expression values overall with a considerable increase in male thorax, female thorax and whole body (NCBI BioProjects PRJNA609978 and PRJNA448935) (Figure 4 and Table 4 in GigaDB [79]).

*PGK* catalyzes the reversible conversion of 1,3-bisphosphoglycerate to 3-phosphoglycerate (3PG) while generating one molecule of ATP in the seventh step of glycolysis. A single gene was annotated in *D. citri*, and other insects also have single copies



**Table 4.** Gene counts in selected insect species.

| Genes | D. citri | A. pisum | T. castaneum | A. mellifera | D. melanogaster |
|---|---|---|---|---|---|
| *Hexokinase (HK)* | 3 | 3 | 2 | 1 | 4 |
| *Phosphoglucose isomerase (PGI)* | 1 | 1 | 1 | 1 | 1 |
| *Glucose-6-phosphate 1-epimerase* | 1 | 1 | 1 | 1 | 1 |
| *ATP Dependent 6-Phosphofructokinase (PFK)* | 1 | 2 | 1 | 1 | 1 |
| *Fructose bisphosphate-aldolase (ALDA or ALDOA)* | 2 | 1 | 1 | 1 | 1 |
| *Triosephosphate isomerase (TPI)* | 1 | 1 | 1 | 1 | 1 |
| *Glyceraldehyde-3-phosphate dehydrogenase (GAPDH)* | 2 | 1 | 2 | 2 | 2 |
| *Phosphoglycerate kinase (PGK)* | 1 | 1 | 1 | 1 | 1 |
| *Phosphoglycerate mutase (PGAM)* | 2 | 1 | 2 | 1 | 2 |
| *Enolase* | 1 | 1 | 3 | 2 | 1 |
| *Pyruvate kinase (PYK)* | 2 | 1 | 4 | 6† | 6† |
| *Pyruvate carboxylase (PC)* | 1 | 1 | 1 | 1 | 1 |
| *Phosphoenolpyruvate carboxykinase (PEPCK)* | 2 | 1 | 1 | 1 | 1 |
| *Phosphoglucomutase 1 (PGM1)* | 1 | 1 | 1 | 1 | 1 |
| *Phosphoglucomutase 2 (PGM2)* | 1 | 1 | 1 | 0 | 2* |
| *Aldose 1-epimerase (GALM)* | 2 | 3 | 2 | 3 | 1 |
| *Fructose 1,6-bisphosphatase (FBPase)* | 1 | 2 | 1 | 2 | 1 |
| *Glucose-6-phosphatase (G6P)* | 0 | 0 | 0 | 0 | 1 |
| *Trehalose-6-phosphate synthase (TPS)* | 2 | 1 | 2 | 1 | 1 |
| *Trehalose-6-phosphate phosphatase (TPP)* | 0 | 0 | 0 | 0 | 0 |
| *Trehalase 1 (TREH-1)* | 1 | 1 | 1 | 1 | 1 |
| *Trehalase 2 (TREH-2)* | 1 | 0 | 1 | 0 | 0 |
| *Trehalose transporter 1 (TRET1)* | 1 | 1 | 1 | 1 | 1 |
| *Trehalose transporter 2 (TRET2)* | 1 | 1 | 1 | 1 | 1 |
| *Glucose transporter (GLUT1)* | 1 | 1 | 1 | 1 | 1 |

The number of genes identified in glycolysis, gluconeogenesis and trehaloneogenesis in *D. citri* and related organisms. †indicates that there are possibly more *PYK* genes in *D. melanogaster* and potentially six in *A. mellifera*. * indicates that there is *phosphoglucomutase 2a* and *2b* in *D. melanogaster*. Copy numbers for the orthologs were obtained from NCBI [56], OrthoDB [67], and Flybase [68].

(Table 4). *PGAM* is an enzyme that converts 3-phosphoglycerate to 2-phosphoglycerate. Members of the *PGAM* family share a common *PGAM* domain, and function as either phosphotransferases or phosphohydrolases [80]. Two copies of *PGAM* were annotated in the *D. citri* genome (Table 4). *PGAM 1* (Dcitr03g11640.1.1) has high expression evident in the midgut and the other paralog, *PGAM 2* (Dcitr03g17850.1.1) is highly expressed in the whole body (Figure 4).

*Enolase* catalyzes the conversion of 2-phosphoglycerate to phosphoenolpyruvate in the ninth step of the glycolytic pathway and a single copy was annotated in the *D. citri* genome (Table 4). RNAi knockdown of the *a-enolase* in *Nilaparvata lugens* reduced egg production, offspring, and hatching rate; however, mortality of adults was unaffected [80]. Pairwise alignment between the *N. lugens* and *D. citri* sequences reveal the characteristics of the *enolase* family: a hydrophobic domain (AAVPSGASTGI) in the N-terminal region at position 31–41, a seven amino acid substrate binding pocket (H159, E211, K345, HRS373-375, and K396), a metal-binding site (S38, D246, E295, and D320) and the *enolase* signature motif (LLLKVNQIGSVTES) [81].

*PYK* catalyzes the irreversible transfer of a phosphoryl group from phosphoenolpyruvate to ADP; thus generating pyruvate and a second ATP molecule, the



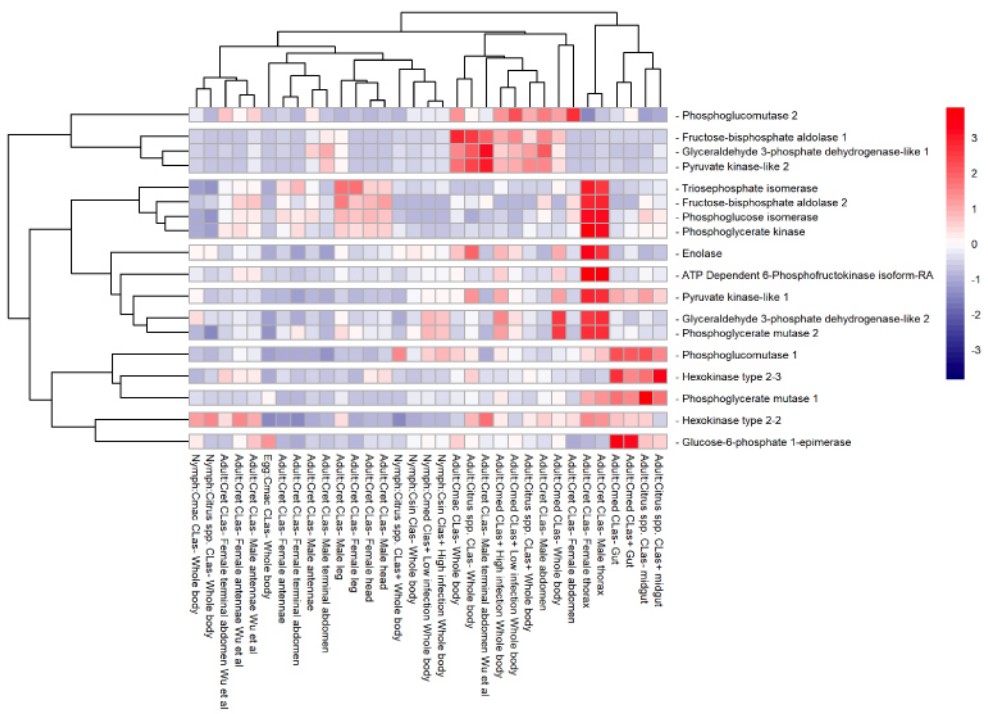

**Figure 4.** Comparison of RNA-seq datasets of genes involved in glycolysis. The heatmap shows results from *D. citri* reared on various citrus varieties, both infected and uninfected with *C*Las. Expression values were collected from CGEN [25]. Data in the heatmap show transcripts per million scaled by gene. RNA-seq data are available from NCBI Bioproject's PRJNA609978 and PRJNA448935 and in addition to several published datasets [9, 72–75]. Expression data for *HK* type 2-1 (Dcitr03g04910.2.1) are not present in the heatmap.

end products of the glycolysis reaction. The copy number of *PYK* varies among insects; *A. mellifera* and *D. melanogaster* both contain six, and *A. gambiae* has only one (Table 4). In *D. citri,* two *PYK* genes were characterized and annotated (Table 2). One of the *PYK* genes (Dcitr07g06140.1.1) is highly expressed in male and female thorax and the other *PYK* gene (Dcitr01g11190.1.1) has relatively low overall expression with the highest expression in the male terminal abdomen (Figure 4). Expression analysis of the enzymes from this phase of glycolysis in thoracic tissue shows that the highest expression is observed for *GAPDH-like 2* and *PYK-like 1* and the lowest occurs for both *GAPDH-like 1* and *PYK-like 2* (Figure 5). In addition, *PGK* (Dcitr00g01740.1.1) and *enolase* (Dcitr02g07600.1.1) also have high expression in the male and female thorax and *PGAM 2* (Dcitr03g17850.1.1) has high expression in whole body besides the male and female thorax (NCBI BioProject PRJNA609978, NCBI BioProject PRJNA448935) (Figure 4 and Table 4 in GigaDB [79]).

## Enzymes of gluconeogenesis

Gluconeogenesis is the metabolic process to re-generate glucose from non-carbohydrate substrates. It uses four specific enzymes. *PC* catalyzes the ATP-dependent carboxylation of pyruvate to oxaloacetate. The curated *PC* model (Dcitr08g01610.1.1) in *D. citri* shows highest overall expression in the male and female thorax, male and female head, and male and female antenna (Figures 6, 7 and Table 5 in GigaDB [79]).

*PEPCK* controls cataplerotic flux and converts oxaloacetate from the tricarboxylic acid cycle to form phosphoenolpyruvate (PEP). Two *PEPCK* genes were annotated and

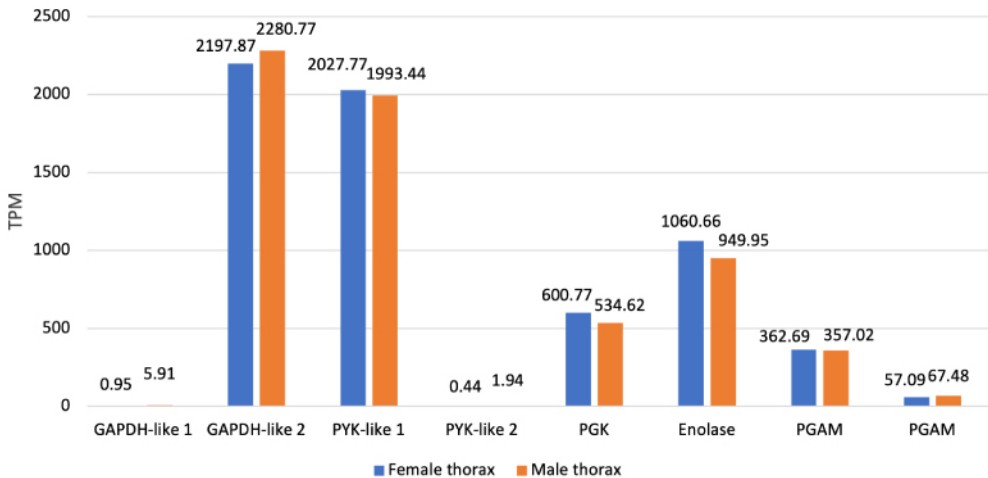

**Figure 5.** Citrus Greening Expression Network expression data for the enzymes involved in the energy production phase in *D. citri*. (GAPDH-like 1: Dcitr10g11030.1.1; GAPDH-like 2: Dcitr01g03200.1.1; PYK-like 1: Dcitr07g06140.1.1; PYK-like 2: Dcitr01g11190.1.1; PGK: Dcitr00g01740.1.1; enolase: Dcitr02g07600.1.1; PGAM: Dcitr03g17850.1.1, Dcitr03g11640.1.1 respectively). Values are based on transcripts taken from the thorax of healthy *C*Las- *D. citri* male and female adults that fed on *C. reticulata*. These experiments had a single replicate. RNA-seq data is available from NCBI BioProject PRJNA448935.

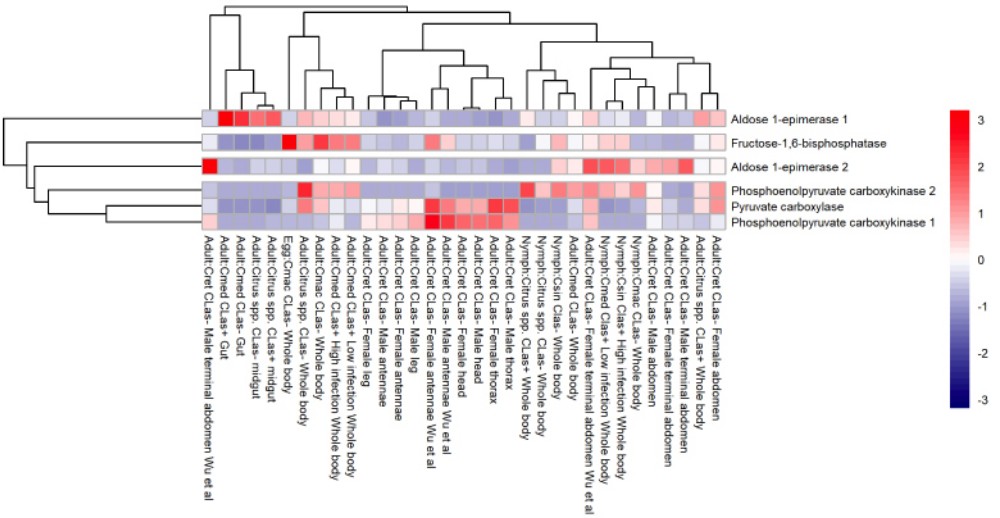

**Figure 6.** Comparison of RNA-seq datasets of genes involved in gluconeogenesis. The heatmap shows results from *D. citri* reared on various citrus varieties, both infected and uninfected with *C*Las. Expression values were collected from the Citrus Greening Expression Network [25]. Data in the heatmap show transcripts per million scaled by gene. RNA-seq data are available from NCBI Bioprojects PRJNA609978 and PRJNA448935 and published datasets [72].

characterized in the *D. citri* genome (Table 2). The first *PEPCK* copy (Dcitr05g10240.1.1) has higher expression in most tissues than all of the other gluconeogenesis genes as is evident in the male and female antenna, male and female thorax, and the male and female head. The highest expression of the second copy of *PEPCK* (Dcitr08g02760.1.1) is shown in the whole body. *FBPase* facilitates one of the three bypass reactions in gluconeogenesis, whereby hydrolysis of fructose-1,6-bisphosphate produces fructose-6-phosphate. A single

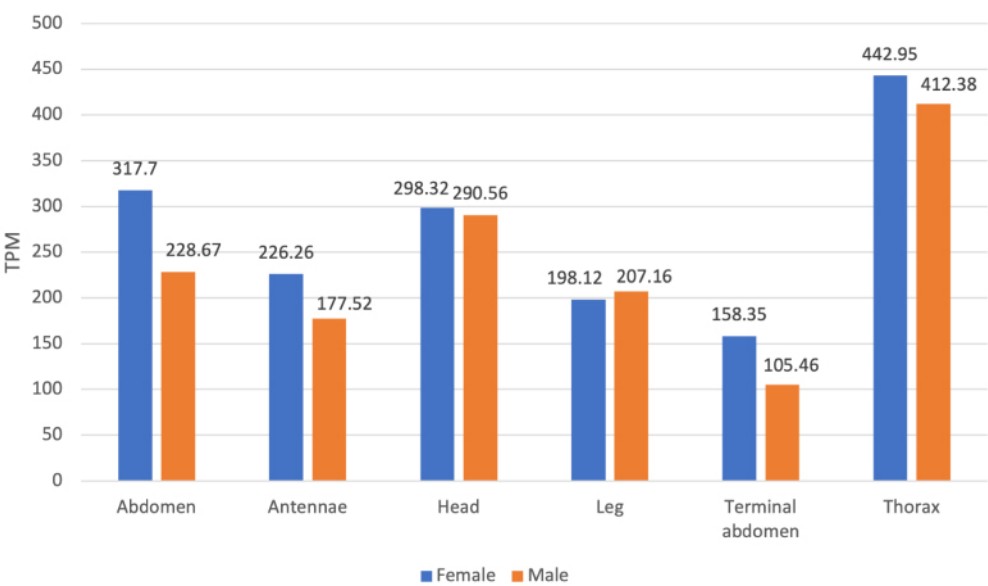

**Figure 7.** Citrus Greening Expression Network expression data of the enzyme pyruvate carboxylase (Dcitr08g01610.1.1) in *D. citri*. Values are based on transcripts isolated from various body parts of healthy *C*Las-*D. citri* adults that fed on *C. reticulata*. These experiments had a single replicate. RNA-seq data is available from NCBI BioProject PRJNA448935.

copy of this gene was annotated in *D. citri,* similar to other insects, although two copies are present in the pea aphid, *A. pisum,* and the honeybee, *A. mellifera* (Table 2). *FBPase* (Dcitr11g08070.1.1) shows highest expression in the egg (Figure 5). *Glucose-6-phosphatase* (*G6Pase* or *G6P*), which is specific to gluconeogenesis, catalyzes the conversion of glucose-6-phosphate to glucose [31]. However, this enzyme is not present in most insect species, including *D. citri.* Though present in *N. lugens*, RNAi studies showed that knockdown of *G6Pase* in *N. lugens* had no effect on the genes involved in trehalose metabolism [82].

## Enzymes of trehaloneogenesis

Trehalose is a non-reducing disaccharide present in many organisms, including yeast, fungi, bacteria, plants and invertebrates. As the main hemolymph sugar in insects, it is found in high concentrations [32, 83]. Trehalose is synthesized from glucose by trehalose-6-phosphate (Tre-6-P), where the mobilization of trehalose to glucose is considered critical for metabolic homeostasis in insect physiology [30]. Synthesis of trehalose occurs in the fat body, when stimulated by neuropeptides from the brain [32]. These peptides decrease the concentration of fructose 2,6-bisphosphate, which strongly activates the glycolytic enzyme *PFK* and inhibits the gluconeogenic enzyme *fructose 1,6-bisphosphatase. Fructose 2,6-bisphosphatase* is thus a key metabolic signal in regulating trehalose synthesis in insects. After synthesis, trehalose is transported through the hemolymph and enters cells through trehalose transporters, where it is converted into glucose by trehalase.

Three enzymes are involved in trehaloneogenesis: *trehalose-6-phosphate synthase* (*TPS*), *trehalose-6-phosphate phosphatase* (*TPP*), and *trehalase* (*TREH*) (Figure 1). *TPS* catalyzes the transfer of glucose from UDP-glucose to G6P forming trehalose 6-phosphate (T6P) and

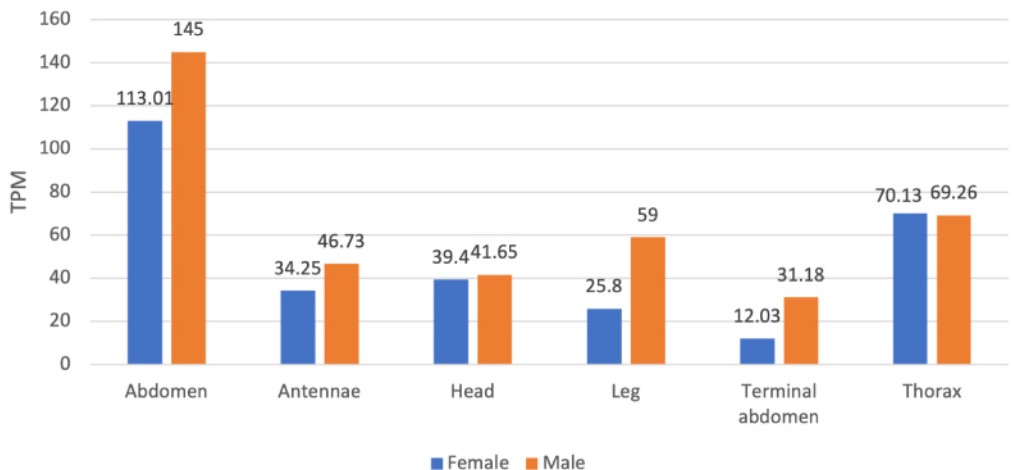

**Figure 8.** Citrus Greening Expression Network expression data of the enzyme trehalose 6-phosphate synthase (Dcitr02g17550.1.1) in *D. citri*. Values are based on transcripts expressed in various body parts of healthy *C*Las- *D. citri* adults that fed on *C. reticulata*. These experiments had a single replicate. RNA-seq data are available from NCBI BioProjects PRJNA448935.

UDP [83]. Targeting a *D. citri TPS* (*DcTPS*) gene for RNAi therapeutics revealed that dsRNA-mediated gene-specific silencing strongly reduced expression of *DcTPS* and survival rate of nymphs, and increased malformation [39]. Two copies of *TPS* were annotated in the v3 genome of *D. citri. TPS 1* (Dcitr02g17550.1.1) had the highest expression, found in the *C*Las+ and *C*Las- adult midgut, respectively (Figure 8 and Table 6 in GigaDB [79]).

In some organisms, *TPP* dephosphorylates T6P to trehalose and inorganic phosphate [84]. However, many insects appear to lack this gene, including *D. citri* as it was not found in the v3 genome. Most insects with multiple *TPS* genes encode proteins with TPS and TPP domains [85, 86]. *TPS* in *Drosophila* appears to have the functions of both *TPS* and *TPP* [87]. *Trehalase* (*TREH*) catalyzes stored trehalose by cleaving it to two glucose molecules. There are two trehalase genes: *TREH-1*, which encodes a soluble enzyme found in hemolymph, goblet cell cavity and egg homogenates, and *TREH-2*, which encodes a membrane-bound enzyme found in flight muscle, ovary, spermatophore, midgut, brain and thoracic ganglia [84]. The two curated *TREH* genes in *D. citri* show different expression in the psyllid. *TREH-1A* (Dcitr07g04030.1.1) shows high expression in the gut and midgut, and *TREH-*2 (Dcitr08g09220.1.1) shows moderate expression in the female thorax and male antennae (Figure 9).

*TREH* is the only enzyme known for the irreversible splitting of trehalose in all insects [84] and *D. citri and T. castaneum* are the only insects with the second copy, *TREH-2* (Table 2).

The two main trehalose transporters are trehalose transporter 1 (TRET1) and trehalose transporter 2 (TRET2), which both transport trehalose to and from cells with *TREH*. One gene copy for each of these trehalose transporters was annotated in *D. citri* (Table 2). Expression analysis shows that *TRET1* (Dcitr01g17710.1.1) is highly expressed in the gut and *TRET2* (Dcitr00g03240.1.1) is moderately expressed in the male abdomen (Figure 9).

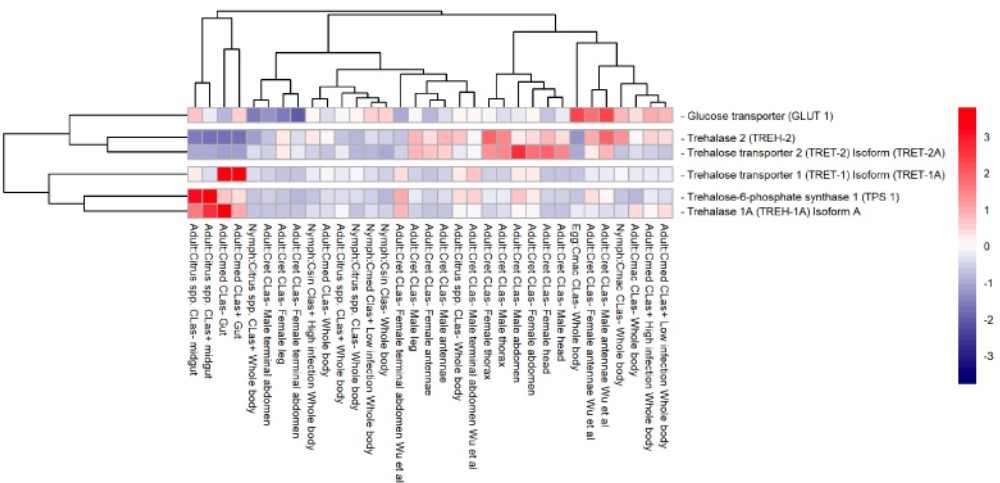

**Figure 9.** Comparison of RNA-seq datasets of genes involved in trehaloneogenesis. The heatmap shows results from *D. citri* reared on various citrus varieties, both infected and uninfected with *C*Las. Expression values were collected from the Citrus Greening Expression Network [25]. Data in the heatmap show transcripts per million scaled by gene. RNA-seq data are available from NCBI Bioprojects PRJNA609978 and PRJNA448935 and published datasets [72]. Expression data for trehalose-6-phosphate synthase 2 (TPS 2), trehalase 2 (TREH-2), trehalose transporter 1 (TRET1) isoform (TRET1B), trehalose transporter 2 (TRET2) isoform (TRET2B), and trehalose transporter 2 (TRET2) isoform (TRET2C) are not present in the heatmap.

## CONCLUSION

Manual annotation of the central metabolic pathways of glycolysis, gluconeogenesis, and trehaloneogenesis provides the accurate gene models required for development of molecular therapeutics to target *D. citri*. RNAi studies targeting genes involved in trehalose metabolism produced significant mortality in *D. citri*, [39, 88], demonstrating the functional application of the genes identified. Expression analysis of the genes annotated in carbohydrate metabolism pathways identified differences related to life stage, sex and tissue. Female insects preferentially feed on diets high in protein and males tend towards carbohydrate-rich diets [89–91]. Gene expression patterns reflect this dietary preference between males and females in *A. gambiae*, where males feeding entirely on sugar have elevated gene expression for carbohydrate metabolism [92]. A similar expression pattern was reported for *N. lugens*, where gene sets related to carbohydrate metabolism were upregulated in males compared with females [93]. Several of the carbohydrate metabolism genes in *D. citri*, including *HK*, *PFK*, *aldolase*, *GAPDH-like 1*, *PGAM 1* and *2*, *PYK*, *enolase*, *PEPCK 1* and *2*, *aldose 1-epimerase*, *TPS 1*, *TREH1A* and *2*, and *TRET1* and *2*, show increased expression in various tissues in males than in females (results in [79]). Annotation of the carbohydrate metabolism genes advances the understanding of the basic biology of *D. citri* and will aid in the development of RNAi-based applications.

## REUSE POTENTIAL

The manually curated gene models were annotated through a collaborative community project [11] to further understand psyllid biology and with a goal to annotate gene families related to immune response, metabolism and other major functions [94]. Continued examination of the glycolysis, gluconeogenesis, and trehaloneogenesis pathways across arthropods, and especially in insect vectors like *D. citri*, will provide novel and

species-specific gene targets to control psyllid populations (potentially through RNAi) and reduce the effects of pathogens such as *C*Las.

## DATA AVAILABILITY

The datasets supporting this article are available in the *GigaScience* GigaDB repository [79]. The gene models are part of an updated official gene set (OGS) for *D. citri* submitted to NCBI under Bioproject PRJNA29447. The OGS (v3) is also publicly available for download, BLAST analysis and expression profiling on Citrusgreening.org and the Citrus Greening Expression Network [25]. The *D. citri* genome assembly (v3), OGS (v3) and transcriptomes are accessible on the Citrusgreening.org portal [12]. Accession numbers for genes used in multiple alignments or phylogenetic trees are provided in Table 1.

## EDITOR'S NOTE

This article is one of a series of Data Releases crediting the outputs of a student-focused and community-driven manual annotation project curating gene models and, if required, correcting assembly anomalies, for the *Diaphorina citri* genome project [95].

## DECLARATIONS
## LIST OF ABBREVIATIONS

ADP: adenosine diphosphate; *Am*: *Apis mellifera*; *Ap*: *Acyrthosiphon pisum*; ATP: adenosine triphosphate; CGEN: Citrus Greening Expression Network; *C*Las: *Candidatus* Liberibacter asiaticus; *Cmac*: *Citrus macrophylla; Cmed*: *Citrus medica*; *Cret*: *Citrus reticulata*; *Csin*: *Citrus sinensis*; *Dc*: *Diaphorina citri*; DHAP: dihydroxyacetone phosphate; *Dm*: *Drosophila melanogaster*; *FBPase*: *fructose-1,6-bisphosphatase*; GAP: glyceraldehyde-3-phosphate; *GAPDH*: *glyceraldehyde 3-phosphate dehydrogenase*; *G6Pase/G6P*: *glucose-6-phosphatase*; *Hh*: *Halyomorpha halys*; *HK*: *hexokinase*; Iso-seq: Isoform sequencing; MCOT: Maker, Cufflinks, Oases, Trinity; NCBI: National Center for Biotechnology Information; *Nl*: *Nilaparvata lugens*; *PC*: *pyruvate carboxylase*; *PEPCK*: *phosphoenolpyruvate carboxykinase*; *PFK*: *phosphofructokinase*; *PGAM*: *phosphoglycerate mutase*; *PGI*: *phosphoglucose isomerase*; *PGK*: *phosphoglycerate kinase*; *PYK*: *pyruvate kinase*; RNAi: RNA interference; RNA-seq: RNA sequencing; *Tc*: *Tribolium castaneum*; *TPI*: *triosephosphate isomerase*; TPM: transcripts per million; *TPP*: *trehalose-6-phosphate phosphatase*; *TPS*: *trehalose-6-phosphate synthase*; TRE: trehalose; T6P: trehalose-6-phosphate.

## ETHICAL APPROVAL

Not applicable.

## CONSENT FOR PUBLICATION

Not applicable.

## COMPETING INTERESTS

The authors declare that they have no competing interests.

## AUTHORS' CONTRIBUTIONS

WBH, SJB, TD and LAM conceptualized the study; TD, SS, TDS and SJB supervised the study; SJB, TD, SS and LAM contributed to project administration; BT, AM, KK, CV, CM, DH and SA conducted investigation; PH, MF-G and SS contributed to software development; SS, TDS, JB

PH and MF-G, developed methodology; SJB, TD, WBH and LAM acquired funding; BT, DLH, MRJ, AM and KK prepared and wrote the original draft; TD, SJB, SS, NP, TDS, WBH and JB reviewed and edited the draft.

## FUNDING

This work was supported by USDA-NIFA grants 2015-70016-23028, HSI 2020-38422-32252 and 2020-70029-33199.

## ACKNOWLEDGEMENTS

We would like to thank Helen Wiersma-Koch (Indian River State College) and Thomson Paris (USDA-ARS-Horticultural Research Laboratories) for assistance.

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
