## [Reviewer Report]

Comments on revised manuscriptThe authors have improved the MS to make it clearer, and I think now the manuscript is suitable for publication in the journal GigaByte.

---

## [Reviewer Report]

Reviewer name and names of any other individual's who aided in reviewer Mary Ann TuliDo you understand and agree to our policy of having open and named reviews, and having your review included with the published papers. (If no, please inform the editor that you cannot review this manuscript.)YesIs the language of sufficient quality?YesPlease add additional comments on language quality to clarify if needed
NAAre all data available and do they match the descriptions in the paper? NoAdditional CommentsThe paper states "The D. citri genome assembly (v3), OGS (v3) and transcriptomes are accessible on the Citrusgreening.org portal"
I believe v2 is available, not v3 yet. Are the data and metadata consistent with relevant minimum information or reporting standards? See GigaDB checklists for examples <a href="http://gigadb.org/site/guide" target="_blank">http://gigadb.org/site/guide</a>YesAdditional CommentsIs the data acquisition clear, complete and methodologically sound?YesAdditional CommentsIs there sufficient detail in the methods and data-processing steps to allow reproduction?YesAdditional CommentsIs there sufficient data validation and statistical analyses of data quality? YesAdditional CommentsIs the validation suitable for this type of data?YesAdditional CommentsIs there sufficient information for others to reuse this dataset or integrate it with other data?NoAdditional CommentsThe paper states "The gene models will be part of an updated official gene set (OGS) for D. citri that will be submitted to NCBI."
Until these models are available in NCBI their reuse is limited. Any Additional Overall Comments to the AuthorRecommendationMinor Revision

---

## [Reviewer Report]

Reviewer name and names of any other individual's who aided in reviewer Xinyu LiDo you understand and agree to our policy of having open and named reviews, and having your review included with the published papers. (If no, please inform the editor that you cannot review this manuscript.)YesIs the language of sufficient quality?YesPlease add additional comments on language quality to clarify if needed
Are all data available and do they match the descriptions in the paper? YesAdditional CommentsAre the data and metadata consistent with relevant minimum information or reporting standards? See GigaDB checklists for examples <a href="http://gigadb.org/site/guide" target="_blank">http://gigadb.org/site/guide</a>YesAdditional CommentsIs the data acquisition clear, complete and methodologically sound?YesAdditional CommentsIs there sufficient detail in the methods and data-processing steps to allow reproduction?YesAdditional CommentsIs there sufficient data validation and statistical analyses of data quality? YesAdditional CommentsIs the validation suitable for this type of data?YesAdditional CommentsIs there sufficient information for others to reuse this dataset or integrate it with other data?YesAdditional CommentsAny Additional Overall Comments to the AuthorIn the paper entitled “Annotation of glycolysis, gluconeogenesis, and trehaloneogenesis pathways provide insight into carbohydrate metabolism in the Asian citrus psyllid”, the authors conducted a high quality annotation of genes involved in glycolysis, gluconeogenesis, and trehaloneogenesis in Diaphorina citri genome, which provided the bases to develop gene-targeting therapeutics for this important pest species. 

The MS is well-written, and the analyses are clear and proper. I found some minor concerns that should be addressed. 

In the first paragraph of Page 10, the authors used cross symbol and the asterisk in the sentence “The number of genes identified in glycolysis….from NCBI, OrthoDB, and Flybase.”. However, the cross symbol and the asterisk are used without any explanation or citation. I suggest to cite the Appendix the authors referred to or add an explanation to make it clearer.

In Conclusion part, on Page 15, the authors stated “Expression analysis of the genes annotated in the carbohydrate metabolism pathways identified differences related to life stage, sex and tissue.”. But what are the differences are not mentioned here. I think it would be better to summarize the key/predominant differences about gene expression in the carbohydrate metabolism pathways.

In addition, it is interesting that the gene expression related with carbohydrate metabolism is sexually different in the Asian citrus psyllid. Is it common in insects or existed in some specific groups?RecommendationMinor Revision